# CryoEM structures of open dimers of gyrase A in complex with DNA illuminate mechanism of strand passage

Katarzyna M Soczek[1], Tim Grant[2†], Peter B Rosenthal[2,3], Alfonso Mondragón[1]*

[1]Department of Molecular Biosciences, Northwestern University, Evanston, United States; [2]Division of Physical Biochemistry, MRC National Institute for Medical Research, London, United Kingdom; [3]Structural Biology of Cells and Viruses Laboratory, The Francis Crick Institute, London, United Kingdom

**Abstract** Gyrase is a unique type IIA topoisomerase that uses ATP hydrolysis to maintain the negatively supercoiled state of bacterial DNA. In order to perform its function, gyrase undergoes a sequence of conformational changes that consist of concerted gate openings, DNA cleavage, and DNA strand passage events. Structures where the transported DNA molecule (T-segment) is trapped by the A subunit have not been observed. Here we present the cryoEM structures of two oligomeric complexes of open gyrase A dimers and DNA. The protein subunits in these complexes were solved to 4 Å and 5.2 Å resolution. One of the complexes traps a linear DNA molecule, a putative T-segment, which interacts with the open gyrase A dimers in two states, representing steps either prior to or after passage through the DNA-gate. The structures locate the T-segment in important intermediate conformations of the catalytic cycle and provide insights into gyrase-DNA interactions and mechanism.

DOI: https://doi.org/10.7554/eLife.41215.001

*For correspondence:
a-mondragon@northwestern.edu

Present address: †Janelia Research Campus, Howard Hughes Medical Institute, Virginia, United States

Competing interests: The authors declare that no competing interests exist.

## Introduction

DNA topoisomerases are versatile enzymes that modify the topology of DNA and are present in all three domains of life (*Schoeffler and Berger, 2008*). They are involved in many cellular processes and help solve problems associated with DNA manipulations (*Schoeffler and Berger, 2008*). For example, they are responsible for preventing unwanted DNA overwinding in front of the replication and transcription forks as well as for unknotting of copied DNA strands behind the replication fork (*Wang, 2002*). In order to modify the topology of DNA, topoisomerases introduce either a transient single stranded (type I enzymes) or a transient double stranded (type II enzymes) break in the phosphodiester backbone of the DNA chain, followed by passing of DNA strand(s) through the break, and final resealing of the phosphodiester backbone. Whereas all type II topoisomerases change the linking number strictly in steps of two, type I enzymes can change the linking number in steps of one (type IA) or any number (type IB and IC). In all cases, breaking the DNA phosphodiester backbone involves a transient phospho-tyrosine bond. All members of the same sub-type show structural and sequence similarities, but there are also clear similarities between type IA and type IIA enzymes, as both use an enzyme-bridged strand passage mechanism, break/religate the DNA in similar manner, and show common structural domains (*Schoeffler and Berger, 2008*). Due to their critical role in the cell, topoisomerases are important targets for antibiotics and chemotherapeutics (*Collin et al., 2011*; *Pommier et al., 2010*).

DNA gyrases are type IIA topoisomerases found in bacteria, archaea, and some eukaryotes (plants [*Evans-Roberts et al., 2016*] and plasmodial parasites [*Dar et al., 2007*]) and are capable of relaxing positive supercoils and introducing negative supercoils into DNA as well as performing

other topological manipulations (*Reece and Maxwell, 1991b*). Unlike eukaryotic type IIA enzymes, which are large homodimers, gyrase is an $A_2B_2$ heterotetramer formed by two GyrA and two GyrB subunits (*Mizuuchi et al., 1978*). Introduction of negative supercoils into DNA is a unique function of gyrase. It is coupled to ATP hydrolysis and requires the C-terminal domains (CTDs) of the GyrA subunits, which are involved in binding and guiding the DNA through the negative supercoiling cycle (*Lanz and Klostermeier, 2011*; *Lanz and Klostermeier, 2012*; *Reece and Maxwell, 1991a*) and are not present in other type IIA topoisomerases. Removal of the GyrA CTD abolishes the ability of gyrase to negatively supercoil DNA, but the truncated enzyme can relax both negative and positive supercoils, similarly to other type IIA topoisomerases (*Kampranis and Maxwell, 1996*). The mechanism of action employed by type IIA enzymes in general, and gyrase in particular, has been extensively studied by a variety of techniques (*Basu et al., 2016*), and a picture of the steps involved in changing the topology of DNA has emerged (for example [*Gubaev and Klostermeier, 2014*; *Kampranis et al., 1999*]). The mechanism involves the concerted breakage of the double stranded DNA backbone and the formation of a protein-mediated DNA opening (DNA-gate) followed by passage of another DNA strand through the gate, religation of the DNA phosphodiester backbone, and release of the passed stand through a second protein gate (C-gate) (*Figure 1*). In the proposed mechanism, a series of large conformational changes in the proteins, movements of the DNA strands, as well as opening and closing of gates in the DNA and the protein, need to occur in a concerted fashion (*Figure 1*). The GyrA dimer forms two gates that facilitate DNA passage. One gate (DNA gate) binds DNA (G-segment) at the beginning of the cycle and cleaves the DNA using its conserved, catalytic tyrosines. The second gate (C-gate) opens later in the cycle to release the transported DNA segment (T-segment). GyrB contains an ATPase domain and the GyrB dimer forms the third of the gyrase gates (N-gate).

Our understanding of gyrase global subunit arrangement and relative GyrA and GyrB positions comes from low resolution EM structures (*Papillon et al., 2013*). Atomic structural information exists at the level of truncated subunits and full length eukaryotic homologs that were obtained through X-ray crystallography (*Dong and Berger, 2007*; *Fu et al., 2009*; *Laponogov et al., 2018*; *Morais Cabral et al., 1997*; *Ruthenburg et al., 2005*; *Schmidt et al., 2012*; *Wigley et al., 1991*). The most commonly crystallized gyrase fragment consists of a GyrA dimer with truncated CTDs (GyrA-ΔCTD), the C-terminal region of GyrB, and a short G-segment DNA bound and stabilized by a quinolone antibiotic (for example *Bax et al., 2010* and *Blower et al., 2016*). These structures show a conformation of the GyrA dimer, where the

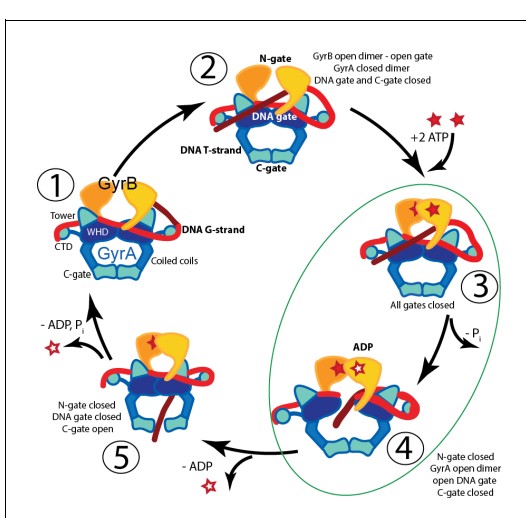

**Figure 1.** Proposed mechanism of supercoiling of DNA by gyrase. The schematic diagram illustrates a simplified model for the proposed mechanism of negative supercoiling by gyrase. (**1**) Gyrase binds a DNA segment (G-segment, red) at the interface of gyrase A (GyrA, blue/cyan) and gyrase B (GyrB, orange) subunits in the region forming the DNA gate. The DNA is wrapped around the C-terminal domains (CTDs) of GyrA. The CTDs are unique to gyrase and are essential for introducing negative supercoils into DNA. (**2**) The CTDs guide a DNA segment (T-segment, dark red) to enter the space in between two GyrB subunits, which also form the N-gate. (**3**) Upon binding of 2 ATP molecules to GyrB, the N-gate closes, trapping the T-segment inside the GyrB subunits. (**4**) The G-segment is cleaved by the active site tyrosines located in the GyrA subunits, forming a covalent protein/DNA intermediate and leading to the opening of the DNA-gate that allows T-segment passage through this gate. One ATP is hydrolyzed at this step, leaving 1 ATP and 1 ADP molecule bound to GyrB. (**5**) After T-segment passage through the gate, the DNA-gate closes, ADP leaves the complex, and the third gate, the C-gate, opens to release the trapped T-segment. Finally, the second ATP is hydrolyzed, the C-gate closes, the N-gate opens, and the enzyme is ready for the next cycle. Steps 3 and 4 are highlighted as they correspond to the states where the complexes of GyrA with DNA provide novel information on T-segment binding and passage. The diagram is based on *Basu et al. (2012)*.
DOI: https://doi.org/10.7554/eLife.41215.002

DNA-gate is closed and the G-segment may be cleaved and poised to be opened. Open conformations of a GyrA truncated dimer with different extents of DNA-gate opening have been observed (*Rudolph and Klostermeier, 2013*) in the absence of DNA. Studies of gyrase homologs have provided structures of open conformations also in the absence of DNA (*Berger et al., 1996*; *Corbett et al., 2005*; *Fass et al., 1999*). Recently, a structure of a fragment of human topoisomerase II spanning the region corresponding to the GyrA-ΔCTD and the TOPRIM domain of GyrB in complex with G-segment DNA was also reported (*Chen et al., 2018*). A structure of GyrB in complex with a putative T-segment has been reported recently (*Laponogov et al., 2018*), but structures where GyrA or the equivalent region in a type II topoisomerase interact with the T-segment have not been observed, even though they are critical to understand the catalytic cycle. The absence of these structures is probably due to the transient nature of the intermediates.

Here we present two different oligomeric structures of *Streptococcus pneumoniae* GyrA-ΔCTD forming an open dimer complex with a 44 bp unbroken DNA oligonucleotide. One of the oligomers is formed by four dimers assembled around linear B-DNA. The protein subunits in this oligomer are arranged with D2 symmetry, however, trapped DNA breaks the overall symmetry of the complex. In this complex, the GyrA-ΔCTD open dimers make two types of interactions with DNA, mimicking a T-segment either prior to entering or just after passing through the DNA-gate. The second complex has tetrahedral symmetry and is composed of six open dimers that trap a tightly bent DNA inside the complex. The DNA acquires multiple conformations, which breaks the overall symmetry of this complex. In both structures the DNA is interacting with positively charged protein regions close to the DNA-gate. The structures correspond to and support protein and DNA orientations predicted (*Chen et al., 2018*; *Gubaev and Klostermeier, 2014*; *Kampranis et al., 1999*), but not previously observed, that are part of the catalytic cycle. The complexes of GyrA with the T-segment DNA provide new insights into conformational states of a type II topoisomerase and DNA in the catalytic cycle.

## Results

### Complex formation and initial characterization

Previously, reconstitution of intact gyrase by mixing GyrA, GyrB, and a 137 bp DNA fragment in the presence of nalidixic acid, which stabilizes the broken G-segment DNA in the closed DNA-gate, enabled the study of the structure of the complex in solution (*Baker et al., 2011*). A limitation of these studies was the mobility of CTDs, even in complex with DNA. To overcome this problem, reconstitution was performed with a GyrA-ΔCTD construct (residues 1–487), GyrB, and a 44 bp DNA fragment, a length of DNA chosen based on previous structures where oligonucleotides ranging from 24 to 34 base pairs formed stable complexes (*Blower et al., 2016*; *Dong and Berger, 2007*). Instead of obtaining a core DNA/gyrase complex analogous in structure to eukaryotic topoisomerase II, we obtained two distinct oligomeric complexes. One of the complexes is characterized by the arrangement of GyrA monomers with tetrahedral symmetry and will be referred to as the 'Tetrahedral complex'. The other complex is built by GyrA dimers arranged with dihedral symmetry and will be referred to as the 'Dihedral complex'.

The Dihedral complex, built from four GyrA-ΔCTD open dimers, was obtained by mixing GyrA-ΔCTD with 44 bp oligonucleotide and ciprofloxacin. In order to characterize this complex by electron microscopy (EM), it was purified on a glycerol/glutaraldehyde gradient using the GraFix procedure (*Kastner et al., 2008*), which stabilizes the complex through mild crosslinking. Negative stain EM data collection followed by 2D classification of the particles showed a complex that was larger than expected based on the size of the components, suggesting possible oligomerization of the subunits. The Tetrahedral complex, built from six GyrA-ΔCTD open dimers, was formed by mixing GyrA-ΔCTD, GyrB, and the oligonucleotide in the presence of ciprofloxacin and novobiocin. This complex was purified and characterized using the same procedures as the Dihedral complex. 2D classification of particles obtained from negative stain EM images showed well-defined classes that in their shapes and sizes did not resemble known topoisomerase II structures and were clearly different from the Dihedral complex. It was observed that when mixing GyrA-ΔCTD with GyrB and DNA both types of complexes can be formed. However, when GyrB is excluded from the mixture, predominantly the Dihedral complex is obtained, suggesting that GyrB facilitates the formation of the

Tetrahedral complex. The conditions for assembly of both complexes were additionally tested by including either one or both antibiotics, with or without GyrB, and in the presence or absence of glutaraldehyde. Importantly, the complexes formed in the presence or absence of glutaraldehyde and the antibiotics (*Figure 2—figure supplement 1A*), which shows conclusively that the oligomers do not require crosslinking or antibiotics for formation.

## Structure of a Dihedral GyrA-ΔCTD-DNA complex

Initial negative stain analysis showed that the protein formed a complex with dihedral symmetry (D2). CryoEM analysis showed that the complex is formed by four dimers of GyrA-ΔCTD in the open conformation and linear DNA threading through the dimers. Further analysis of these maps indicated that the DNA breaks the D2 symmetry of the complex. Maps calculated with the DNA subtracted and with D2 symmetry achieved a higher resolution for the protein parts of the assembly (5.2 Å) (*Figure 2* and *Figure 2—figure supplement 2*, *Figure 2—figure supplement 3*). Furthermore, maps calculated without masking the DNA and with C1 symmetry extended to 7 Å resolution and showed a single, B-DNA molecule running across the center of the complex. The C1 map displayed C2 symmetry and further sub-classification imposing C2 symmetry separated particles with either low DNA occupancy or showing 2 pieces of half-length DNA from particles with fully occupied DNA (*Figure 2—figure supplement 6*). Maps calculated imposing C2 symmetry for the fully occupied particles extended to 6.35 Å resolution (*Figure 2* and *Figure 2—figure supplement 3*). At this resolution, the DNA was easily built as the density showed major and minor grooves, and even bumps for the phosphates in the backbone (*Figure 1C*). In addition, a low resolution cryoEM reconstruction of the Dihedral complex in the absence of antibiotics or crosslinkers shows the presence of DNA with the same path, further confirming that the DNA in the complex is not a result of crosslinking. (*Figure 2—figure supplement 1B*).

To build a model into the cryoEM density, initially we performed flexible fitting of a GyrA-ΔCTD monomer from a *S. pneumoniae* crystal structure of a closed dimer of GyrA-ΔCTD with a C-terminal fragment of GyrB and G-segment DNA (PDB ID 4Z2C) into the Tetrahedral map and used this model for further rigid body and flexible fitting into the Dihedral map, as described in the Materials and methods section. In order to model all other protein subunits in the assembly, symmetry operations were applied to the dimer, which is the asymmetric unit in the Dihedral complex. Further model refinement was performed as described in the Materials and methods section and resulted in a very good fit to the density. The Dihedral complex consists of four dimers and a single DNA double helix running along the diagonal. The GyrA-ΔCTD monomer is composed of five domains: C-gate, coiled coil, tower, WHD and N-terminal tail (*Figure 3*). The C-gate and WHD domains are built by short helices connected by flexible loops. The WHD domains form the DNA-gate and contain the catalytic tyrosines that cleave double stranded DNA. The coiled coil domains are formed by helices that connect the C-gate and WHD domains, whereas the tower domain sits adjacent to the WHD domain and is built by β-sheets and helices connected by loops. The tower domain helps stabilize the G-segment of DNA in the closed conformation of the GyrA dimer (*Dong and Berger, 2007*). Finally, the N-terminal tail is formed by helical and flexible regions that mediate interactions between GyrA and GyrB. The interactions between dimers in the Dihedral complex are mediated mainly by contacts between the N-terminal tail of a monomer with the N-terminal tails and WHD domains of adjacent dimers as well as WHD-WHD and WHD-tower domain interactions. Each dimer interacts with DNA trapped inside the complex and those interactions are mediated by arginines and lysines from the tower, WHD, C-gate domains as well as the coiled coil close to the C-gate.

The positioning of the DNA across the center of the protein cage results in two distinct protein/DNA interactions. One interaction, hereafter State B, has the DNA above the DNA gate and may correspond to a T-segment just before entering the DNA gate. In the second complex, hereafter State A, the DNA is positioned just below the DNA gate and may correspond to a T-segment after passage through the gate. The structures confirm that the opening of the GyrA dimer is wide enough to accommodate a double stranded DNA helix as an intermediate state in the supercoiling cycle.

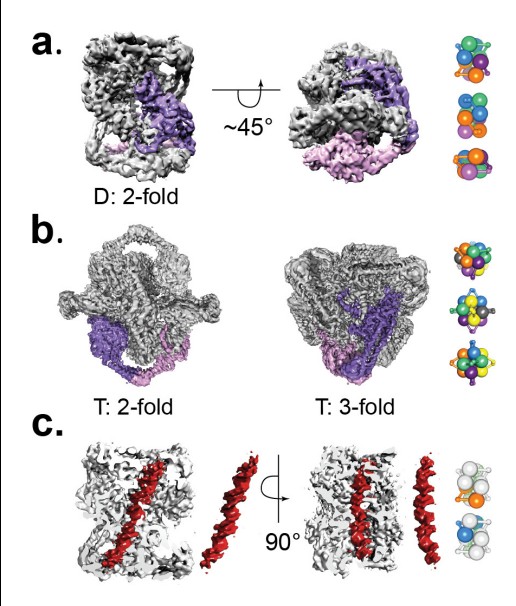

**Figure 2.** Cryo-EM volumes of the Tetrahedral and Dihedral complexes. (a) Diagram showing the 5.2 Å resolution reconstruction of the Dihedral complex with imposed D2 symmetry. The complex is built from four open GyrA dimers. In the diagram, one of the dimers is shown in purple and pink. The dimerization interface of each dimer is formed by the C-gates, as with the Tetrahedral complex. Other interactions between the dimers are mediated by contacts between the N-terminal tails as well as between the N-terminal tails and WHD domains of different dimers and WHD-WHD and WHD-tower domain interactions. On the right, a cartoon depicting the arrangement of the four dimers is shown with views along the two-fold axes (top to bottom). (b) Diagram showing the 4 Å resolution reconstruction of the Tetrahedral complex with imposed tetrahedral symmetry. Twelve GyrA monomers form the complex, arranged as six GyrA open dimers. Two of the monomers forming one of the dimers are shown in purple and pink. The dimerization interface is formed by C-gate interactions. The other interactions between dimers are centered on the N-terminal tails of different monomers, the N-terminal tails with adjacent WHD and tower domains of different monomers as well as WHD-tower domain interactions. On the right, a cartoon depicting the arrangement of the six dimers is shown with views along the three-fold (top) and two fold axes (middle, bottom). Each dimer is shown in a different color. (c) Diagram showing a sliced Dihedral complex volume solved to 6.35 Å and with imposed C2 symmetry. The slice allows the visualization of the 44 bp DNA molecule (red) located in the Dihedral complex interior. The DNA – protein interactions result in two different conformational states of GyrA and DNA. The DNA interacts with the different dimers through positively charged residues in the tower, WHD, C-gate, and the C-gate-adjacent coiled-coil domains. Segmented density for only the DNA

*Figure 2 continued on next page*

## Structure of a Tetrahedral GyrA-ΔCTD assembly at 4.0 Å resolution

A negative stain 3D reconstruction of the complex formed in the presence of GyrB indicated that this complex is comprised of six open GyrA-ΔCTD dimers assembled with tetrahedral symmetry, but surprisingly no GyrB was present. To obtain high-resolution structures, additional cryoEM data sets were collected. Collecting tilted and untilted particles (673,694 particles), imposing tetrahedral symmetry, and masking out the central density improved the resolution to 4.0 Å (*Figure 2A* and *Figure 2—figure supplement 2*, *Figure 2—figure supplement 3*, *Figure 2—figure supplement 4*). When the same data set was analyzed without imposing tetrahedral symmetry on the protein or masking out the interior, the remaining central density could be identified as DNA forming a toroid enclosed by the protein subunits. In order to obtain a more uniform density for the DNA, sub-classification of the 3D classes containing the central density were performed (*Figure 2—figure supplement 4*). Each of the sub-classes was obtained from 2,000 to 3,000 particles and reconstructed without any symmetry, which resulted in a much lower final resolution. Although at lower resolution, two sub-classes with the most prominent central density, features consistent with a DNA molecule, but with slightly different orientation of the DNA, were selected for model building. Two different, but related models of DNA were built based on these densities using flexible fitting of linear B-DNA. Comparison of these models reveals the similarity in the overall shape despite the presence of several DNA orientations in the complex (*Figure 2—figure supplement 5*).

The final map consists of 12 GyrA-ΔCTD monomers related to one another by tetrahedral symmetry. The conformation of GyrA-ΔCTD dimers in this complex is similar to the conformation of the widely open GyrA-ΔCTD dimer observed previously in the crystal structure of *Bacillus subtilis* GyrA-ΔCTD (*Rudolph and Klostermeier, 2013*) (PDB ID 4DDQ). A model was built into the cryoEM map by flexible fitting of a GyrA-ΔCTD monomer from a *S. pneumoniae* crystal structure of a closed dimer of GyrA-ΔCTD with C-terminal fragment of GyrB and G-segment DNA (PDB ID 4Z2C), as described in the Materials and methods section. Subsequent refinement provided a very good fit into the density. In the assembly, GyrA dimers are situated at the two-fold axes of the complex and make

*Figure 2 continued*

(red) is shown. On the right, cartoons depicting the two dimers that interact differently with DNA are shown. The top cartoon corresponds to the GyrA dimer with the DNA positioned below the DNA-gate (orange), while the bottom cartoon shows the GyrA dimer with the DNA positioned above the DNA-gate (blue).

DOI: https://doi.org/10.7554/eLife.41215.003

The following figure supplements are available for figure 2:

**Figure supplement 1.** The Dihedral complexes do not require antibiotics or cross-linkers for assembly.

DOI: https://doi.org/10.7554/eLife.41215.004

**Figure supplement 2.** Reconstructed volumes for the Tetrahedral and Dihedral complexes.

DOI: https://doi.org/10.7554/eLife.41215.005

**Figure supplement 3.** FSC curves for the final structures.

DOI: https://doi.org/10.7554/eLife.41215.006

**Figure supplement 4.** Workflow diagram for the Tetrahedral GyrA reconstruction.

DOI: https://doi.org/10.7554/eLife.41215.007

**Figure supplement 5.** DNA in the Tetrahedral complex.

DOI: https://doi.org/10.7554/eLife.41215.008

**Figure supplement 6.** Workflow diagram for the Dihedral GyrA reconstruction.

DOI: https://doi.org/10.7554/eLife.41215.009

**Figure supplement 7.** Negative stain reconstructions of the Dihedral and Tetrahedral complexes.

DOI: https://doi.org/10.7554/eLife.41215.010

**Figure supplement 8.** Representative 2D classes for the Tetrahedral and Dihedral Complexes.

DOI: https://doi.org/10.7554/eLife.41215.011

contacts at the C-gate domains, whereas the interactions between neighboring dimers at the three-fold axes involve contacts between the N-terminal tails as well as interactions between the N-terminal tails and adjacent tower and WHD domains. In addition WHD and tower domains interact with each other. In all models where no symmetry was imposed to visualize the DNA, the interactions between the protein and the DNA are mainly mediated by residues in the vicinity of the catalytic tyrosine as well as arginines in the tower domain.

## Conformational changes and structural flexibility of the open GyrA-ΔCTD dimer

Comparisons of the open dimers from the Tetrahedral and Dihedral complexes with the closed dimer from the crystal structure of *S. pneumoniae* dimer (PDB ID 4Z2C) show rearrangements of the protein domains and subunits (*Figure 3* and *Figure 3—figure supplement 1*). Additional superpositions of the dimers from the Dihedral and Tetrahedral complexes on each other (*Figure 3* and *Figure 3—figure supplement 1*) as well on the *B. subtilis* open dimer (*Rudolph and Klostermeier, 2013*) (*Figure 3—figure supplement 1*) illustrate the structural flexibility of the open state. Interestingly, whereas the Tetrahedral dimer monomers are related to each other by the point group symmetry, the Dihedral dimer monomers are not related by the point group symmetry and hence are different (*Figure 3—figure supplement 1*). The comparisons show that every domain in the protein changes its relative position, which contributes to the overall conformational change. Overlaying the dimers from the Dihedral and Tetrahedral complexes on the structure from *B. subtilis* GyrA shows the extent of the flexibility of the protein and that the opening can be achieved in slightly different ways. Whereas the coiled coil domains move similarly in all cases, the tower domain moves in opposite direction in the Tetrahedral and Dihedral complexes when compared to the *B. subtilis* GyrA open dimer.

All dimer superpositions show a broad spectrum of GyrA domain movements when going from the closed to the open conformations, as well as a significant variability in the open dimer conformation. It is apparent that the transition from the closed to the open state requires relative rearrangements of every GyrA domain and not a unidirectional movement by a rigid monomer. For example, detailed comparison of the Tetrahedral dimer with the closed dimer shows that the largest conformational change takes place at the N-terminal tail region and the exit C-gate, leaving the winged helix (WHD domain), tower, and coiled coil domains moderately rearranged. These changes potentially reflect concerted motions that individual GyrA domains undergo when the GyrA dimer opens during the DNA supercoiling cycle. Comparison of the cryoEM open dimers with the *B. subtilis* open dimer show different degrees of gate opening, with the largest for *B. subtilis* and the smallest for the Dihedral dimer (*Figure 3—figure supplement 1*). Interestingly, when all open dimers are compared by superposing one of the monomers, it is clear that the opening of the gate is not done in the same manner. The opening involves a rotation separating the domains and also a tilt roughly perpendicular to it, which means that the farther away from the C-gate a particular domain is, the

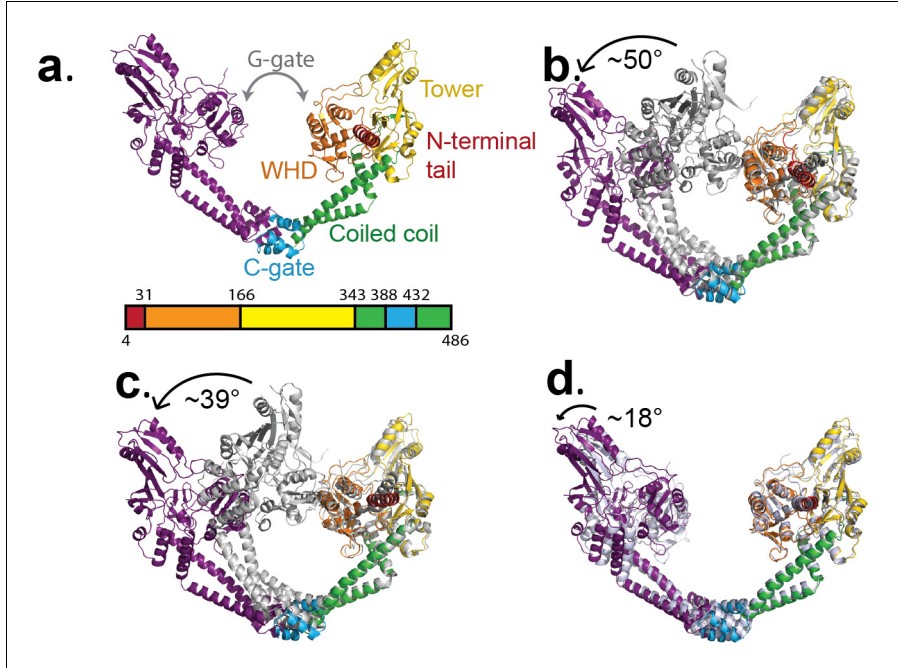

**Figure 3.** Conformational flexibility of GyrA. (**a**) Ribbon diagram of the GyrA dimer with the different domains highlighted in different colors in one of the monomers, the other monomer in solid purple. The bar underneath shows the location of the different domains in the primary sequence colored identically to the figure. (**b**). Ribbon diagram showing the superposition of one of the monomers of the Tetrahedral dimer (color) on a monomer of a closed dimer (PDB ID 4Z2C) (grey) from the same organism. The angle shown represents the magnitude of the rotation needed to rotate the non-superposed monomer to create the open conformation. (**c**). Ribbon diagram showing the superposition of one of the monomers of the Dihedral dimer (color) on a monomer of a closed dimer (PDB ID 4Z2C) (grey) from the same organism. As before, the angle corresponds to the rotation needed to open the closed conformation. (**d**). Ribbon diagram showing the superposition of a monomer of the Dihedral dimer (color) on a monomer of the Tetrahedral dimer (grey). The two open GyrA dimers are very similar. The angle shows the rotation needed to superpose the other pair of monomers. Note that the difference between the Tetrahedral and Dihedral dimers is not simply a rotation around the C-gate, but it involves a rotation centered roughly on the C-gate and also twisting of the WHD and tower domains.
DOI: https://doi.org/10.7554/eLife.41215.012

The following figure supplement is available for figure 3:

**Figure supplement 1.** Conformational plasticity of the GyrA dimers.
DOI: https://doi.org/10.7554/eLife.41215.013

longer the distance of that domain from the equivalent domain in the *B. subtilis* open structure is (*Figure 3—figure supplement 1*).

## Protein-DNA interactions in the GyrA open dimers

The DNA in the Dihedral complex is well defined and forms a B-DNA double helix that threads through the openings in the four dimers forming the complex. The T-segment DNA is not perfectly straight, but instead it curves as it passes through the dimers guided by the gate openings and positively charged lysines and arginines in WHD, tower and C-gate domains. The DNA interacts primarily with three regions in the protein dimer, positively charged patches formed by lysines and arginines in both the WHD and tower domains, a positively charged patch near the C-gate, and the arginines in the neighborhood of the catalytic tyrosine near the DNA-gate. (*Figure 4*). In two of the four dimers the DNA passes through the subunits in an identical manner, resulting in two types of GyrA-ΔCTD/DNA interactions in the oligomer (*Figure 4*). The positioning of the DNA mimics a T-segment in two distinct states during the strand passage (State A and State B). In State A the DNA interacts with arginines and lysines in the WHD and C-gate domains, while in State B, DNA interacts with WHD and tower domains. These DNA-interacting regions are highly conserved (*Figure 4—figure*

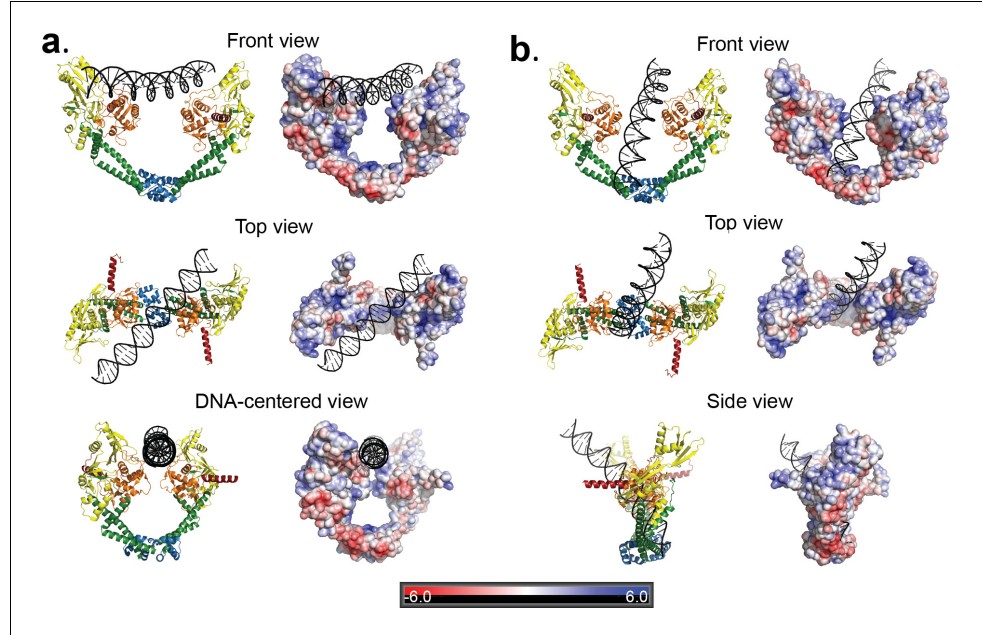

**Figure 4.** Two distinct open GyrA/DNA complexes observed in the Dihedral assembly. The Dihedral complex is built by four GyrA open dimers that show the same protein conformation but that interact with the single DNA molecule in two distinct ways. (**a**) Ribbon diagrams showing the position of the putative T-segment DNA right above the DNA-gate (State B). The position corresponds to the state prior to T-segment passage. Each of the three different views shows a ribbon representation of the complex as well as the electrostatic surface of the protein with a stick diagram of the DNA. The top view illustrates that the DNA runs diagonally above the gate and lies against the positively charged residues in the tower and WHD domains of both monomers. (**b**) Ribbon diagrams showing the position of the putative T-segment DNA after passage through the DNA-gate (State A) in three different views. The front and top views illustrate the DNA interactions with the positively charged residues in the WHD, C-gate, and the adjacent coiled coil domains. The DNA is found inside the protein dimer and it interacts directly with residues in the WHD and also the coiled coil near the C-gate. The electrostatic surface was calculated with APBS (*Baker et al., 2001*) and is rendered with a range of ±6 kT/e. The bar at the bottom corresponds to the color gradient of the electrostatic potential.

DOI: https://doi.org/10.7554/eLife.41215.014

The following figure supplement is available for figure 4:

**Figure supplement 1.** Sequence and charge conservation in the GyrA dimer.

DOI: https://doi.org/10.7554/eLife.41215.015

*supplement 1*), suggesting a universal mode of DNA interaction among all gyrases. Modeling of a G-segment on the open complex shows that the T-segment DNA in State A aligns with the G-segment DNA with an approximately 82° angle and the DNA in State B forms an approximately 64° angle with the G-segment DNA (*Figure 5*). These angles are consistent with computational and experimental predictions for the angles between the G- and T-segments that would be formed by a positive supercoil wrap induced by gyrase prior to strand passage (~60°) (*Chen et al., 2018*; *Stone et al., 2003*) and for angles between the segments in the relaxation of positive supercoils by Topo IV (~85°) (*Neuman et al., 2009*). Superposition of the Tetrahedral dimer on the Dihedral dimer with the DNA above the DNA-gate shows that in the Dihedral complex the WHD and tower domains move to maximize the specific interactions with DNA; the twisting of the domains as they open facilitates this interaction (*Figure 5—figure supplement 1*).

The DNA in two Tetrahedral complex sub-classes analyzed has an overall toroidal shape, but is differently oriented inside the protein cage. Based on these sub-classes, two models with slightly different DNA orientations were built (*Figure 2—figure supplement 5*). Toroids themselves may be superimpositions of bent DNA in different orientations and at the present resolution we cannot interpret their possible role or whether they represent intermediates.

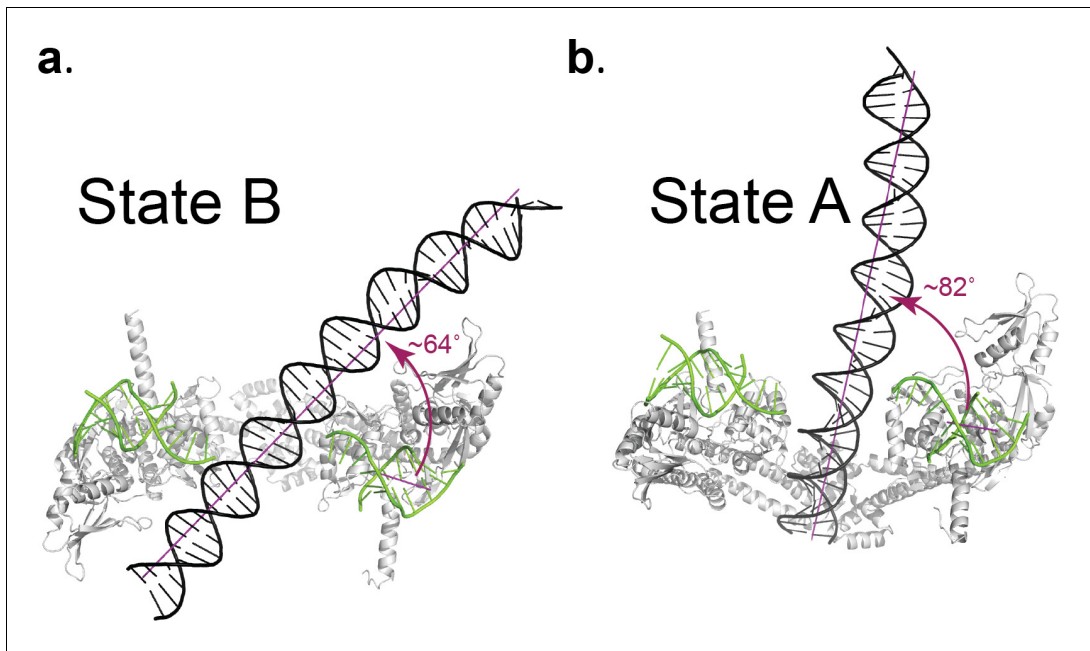

**Figure 5.** Angle between the G- and T-segments in the Dihedral complexes. Models of the GyrA complex with both the G-segment and the T-segment present were built to measure the angle between the G- and T-segments in the models. The split G-segment DNA was built based on the structure of the *S. pneumoniae* GyrA/DNA complex in the closed conformation (PDB ID 4Z2C.) (a) Ribbon diagram of the dimer with the DNA aligned above the DNA-gate corresponding to a putative complex formed by the G- and T-segments prior to gate passage. It has been computationally predicted (*Neuman et al., 2009*; *Stone et al., 2003*) that in a positive supercoil the DNA strands should cross at a ~ 60° angle. In the case of the putative GyrA dimer/G-segment/T-segment complex the angle is about 64°, which is close to the predicted positive supercoil angle. (b) Ribbon diagram of a GyrA dimer with the DNA passing through the DNA-gate. In this model, the angle between the G- and T-segments is about ~82° and is close to the ~85° angle shown to be the preferred angle for Topo IV - DNA in the relaxation reaction (*Neuman et al., 2009*; *Stone et al., 2003*). The angles between G- and T- segments were measured between the helical axes calculated by DSSR in X3DNA package (*Lu and Olson, 2003*; *Lu and Olson, 2008*).
DOI: https://doi.org/10.7554/eLife.41215.016

The following figure supplements are available for figure 5:

**Figure supplement 1.** The Tetrahedral dimer conformation cannot accommodate DNA binding above the G-gate in the same manner as the Dihedral dimer conformation.
DOI: https://doi.org/10.7554/eLife.41215.017

**Figure supplement 2.** The T-segment position in State B in the Dihedral dimer conformation corresponds to the position predicted by several models and is compatible with many available structures.
DOI: https://doi.org/10.7554/eLife.41215.018

Overall, the comparison of all available structures of open GyrA dimers shows different degrees of domain movement associated with the opening of the DNA gate. In the cryoEM structures presented here, DNA is present in both oligomers and the DNA-gate opening is smaller than in the open *B. subtilis* GyrA structure. It is possible that different degrees of domain rearrangement are linked to the presence or sensing of DNA in the oligomers and that the observed changes reflect subunit reorganization during the supercoiling cycle.

## Discussion

The last twenty years have brought significant progress in our understanding of the details of the negative supercoiling mechanism by gyrase through a series of structural studies of individual and truncated subunits, complexes with G-segment DNA, as well as a full length complex with DNA in a closed conformation. However, there are no structures of complexes with the T-segment interacting

with GyrA either before or after crossing the DNA gate. We describe here two oligomeric complexes of GyrA-ΔCTD formed under different assembly conditions. The two complexes consist of open dimers of GyrA-ΔCTD with DNA. Although GyrA is not known to oligomerize as observed here and the presence of oligomers in cells is unlikely, the dimers forming the oligomers represent anticipated conformational states. The oligomers serve as scaffolds to stabilize the protein-DNA assemblies, similarly to complexes trapped in crystal lattices. Hence, the conclusions are based on the dimer/DNA interactions and not on the oligomers. The Tetrahedral complex provides a higher resolution structure of the open GyrA dimer whereas the Dihedral complex provides the structures of two anticipated intermediates in the reaction cycle. Open dimers have been observed before without DNA in a crystal lattice (*Rudolph and Klostermeier, 2013*) and are important intermediates in all proposed mechanisms. GyrA-ΔCTD lacks the C-terminal domain that is essential for DNA supercoiling by gyrase, and this may in principle affect the way GyrA interacts with DNA. However, the C-terminal domain is not present in all type II-A topoisomerases and the position of the T-segment and its interactions with the protein are expected to be the same for all type IIA enzymes. In this regard, the structures may capture a general arrangement common to all type IIA topoisomerases. Furthermore, the observations stemming from the cryoEM structures that the interactions between the dimers and DNA involve highly conserved regions, the charge conservation in these regions, and the excellent agreement of the geometry of the complex with proposed models all suggest strongly that the complexes observed represent structures that mimic intermediates in the reaction. The location of the T-segment in State B of the Dihedral complex is also compatible with the recent structure of a fragment of human topoisomerase II with a G-segment in a (partially) open conformation. This assembly corresponds to GyrA-ΔCTD, the C-terminal portion of GyrB, and G-segment DNA (*Chen et al., 2018*). Indeed, when the State B open dimer is aligned with the human topoisomerase II/DNA structure (*Chen et al., 2018*), (*Figure 5—figure supplement 2*), the T-segment DNA from State B fits very well in the groove between the human topoisomerase II A and B subunits, as proposed in that study (*Chen et al., 2018*). Since the topoisomerase II structure has a narrower DNA-gate opening than the Dihedral open dimers, it is possible that the human topoisomerase II structure corresponds to an earlier state in the conformational pathway than the more widely opened State B conformation, where the DNA can pass the DNA-gate. Finally, the predicted angle of the T-segment in reference to G-segment DNA (*Chen et al., 2018*; *Neuman et al., 2009*; *Stone et al., 2003*) is in agreement with the angle of the T-segment observed in State B of our Dihedral complex. The angle between the T-segment and the G-segment is an important aspect of the mechanism of supercoiling as it relates to the geometry of the positive supercoil wrap of DNA induced by gyrase prior to strand passage.

In the Dihedral complex DNA is trapped in two different intermediate states in the supercoiling reaction, States B and A, or before and after passing through the DNA-gate, which are crucial steps that have not been observed before. In both of these states DNA sits within a region containing many positively charged and conserved amino acids (*Figure 4—figure supplement 1*) and is relatively straight, unlike G-segment DNA, which is tightly bent. The G-segment interacts extensively with the protein, which induces the bend and may be necessary to maintain this bent conformation and form the gate. T-segment DNA does not interact as extensively with the protein and does not need to bend to pass through the gate. The absence of strong protein/T-segment interactions may be needed to allow easy transport of the T-segment through the gate, resulting in a less constrained DNA that adopts locally a linear conformation. In State B, DNA interacts with conserved positively charged arginines and lysines in the tower and WHD domains in both monomers in an almost symmetrical manner. In State A, it interacts with the WHD and C-gate domains in an asymmetric manner. The observed positions of the DNA in the dimer are only possible in the open conformation of the protein. The DNA-gate in the GyrA-ΔCTD dimers is open just enough to accommodate the passing strand, so that the positive electrostatic field surrounding the areas above and below the gate remains proximal and may guide passage. A structure of a human topoisomerase II fragment in complex with G-segment DNA (*Chen et al., 2018*) shows a similar, but significantly smaller, opening of the DNA-gate. Modeling using this structure and also the structure of *S. pneumonia* GyrA/GyrB in complex with DNA (PDB ID 4Z2C) confirm that the presence of the C-terminal region of GyrB would not lead to clashes and is compatible with State B (*Figure 5—figure supplement 2*) and State A. Furthermore, the complex provides confirmation that an open GyrA dimer can accommodate a passing DNA strand during the supercoiling cycle and predicts which DNA and C-gate residues are

involved in charge driven interactions with DNA. Moreover, it is possible that the interactions with the WHD and C-gate domains are coupled; as the DNA passes through the DNA-gate it interacts with the C-gate and triggers the conformational changes that open one gate and close the other. It is important to note that we observe T-segment intermediates in the absence of G-segment or GyrB, which may help guide strand passage in the native enzyme. The absence of the G-segment in the complexes means the T-segment is passing through a gate formed only by protein, rather than DNA held apart by the protein. However, the T-segment in our complexes is compatible with other structures, such as the human topoisomerase II/DNA structure (*Chen et al., 2018*) that include the G-segment and polypeptide regions equivalent to portions of GyrB, supporting our conclusion that the complexes reveal *bona fide* strand passage intermediates.

Importantly, the Dihedral and Tetrahedral complexes can coexist when GyrB is included but the Dihedral complex predominates when GyrB is excluded. Addition of GyrB and novobiocin promoted formation of the Tetrahedral complex with highly bent, heterogeneous DNA inside the protein cage. Although GyrB is not incorporated into the Tetrahedral complex, it must be present in the reaction for complex formation and therefore has a role in complex assembly. It is possible that GyrB participates in stabilizing the bent conformation of DNA in the Tetrahedral complex, which then provides a scaffold for the GyrA-ΔCTD dimers to bind and assemble. Overall, the components of the assembly reaction determine which one of the two GyrA-ΔCTD/DNA complexes is formed, but the exact role of GyrB is not clear. In both cases, it appears that the DNA provides the scaffold to promote the assembly.

Whereas the two structures are different, in both models the protein subunits are arranged in a symmetrical way where the DNA breaks the symmetry of the complex. The two structures show that the opening of the DNA-gate requires conformational rearrangement of all GyrA domains and the magnitude of these changes determines the degree of the DNA-gate opening. The twisting and rotation of the GyrA subunits and domains is in accordance with previous studies of GyrA (*Rudolph and Klostermeier, 2013*) and GyrB (*Stanger et al., 2014*). The latter reports three crystal structures of the GyrB ATPase domain with different substrates and mimicking different conformational states of the ATPase domain during the ATP hydrolysis steps. A comparison between the closed, semi-open, and open conformations shows a rotation of the GyrB subunits (*Stanger et al., 2014*) as they move from the closed to the open conformation. This rotation direction is consistent with a transition from a closed GyrA conformation to the Dihedral open dimer and then to the Tetrahedral open dimer. In the case of *S. cerevisiae* topoisomerase II, three different conformational states of the A subunit opening have been reported (*Berger et al., 1996*; *Fass et al., 1999*; *Schmidt et al., 2012*). One state is part of the full length complex and shows a closed conformation (*Schmidt et al., 2012*) whereas the other two are formed by the C-terminal part of the B subunit with the A subunit and show semi-open (T2M) and open conformations (T2O) (*Berger et al., 1996*; *Fass et al., 1999*). A comparison of the closed conformation of *S. pneumoniae* GyrA (PDB ID 4z2c) with T2M and with T2O shows rotation of the A subunits, which is similar to the observed rotation when the GyrA subunits in the closed, Dihedral dimer, and Tetrahedral dimer are compared. Even though the T2M open A dimer has a narrower opening than the Dihedral complex dimer, the overall rotation directionality is consistent. This supports a conclusion that both open dimer states observed in the cryoEM oligomeric complexes are likely to represent intermediate conformations adopted during strand passage reactions performed, not only by gyrase, but also by other type II topoisomerases.

In a mechanism proposed by *Costenaro et al., 2007* and supported by a GyrB-GyrA fusion structure (*Papillon et al., 2013*), GyrB subunits are crossed over above that DNA gate prior to T-segment passage. If crossed over GyrB subunits were to trap the T-segment DNA in its interior cavity, the orientation of the T-segment DNA would have to be reoriented in order to facilitate its passage through the DNA-gate (*Costenaro et al., 2007*). Such T-segment re-alignment would be easily accomplished by rotation of the GyrB subunits allowing for the opening of the GyrB cavity next to the DNA gate. Since GyrB and GyrA are interacting with each other, the rotation/twisting of one dimer would likely induce the rotation of the other.

Based on our findings, we propose that one possible model for the strand passage mechanism involves G-segment cleavage promoting GyrA and DNA-gate opening in a scissor-like motion with a pivot at the C-gate, consistent with the transition from the *S. pneumoniae* GyrA closed conformation to the open dimer conformation observed in the Dihedral complex. Once the T-segment DNA is

released from the GyrB cavity, it is sensed by the GyrA tower domain and aligns with the positively charged path formed by the tower and WHD domains. The GyrA subunits would rotate to attain the Tetrahedral dimer conformation, forming a positively charged funnel between the WHD domains and allowing the DNA to slide through the DNA-gate towards the C-gate. It is not clear what would promote the movement of the T-segment through the G-gate (*Figure 6*, states 3B to 3C), but one possibility is that as the subunits rotate and the gate changes conformation. This alters the electro-static field around both sides of the gate and promotes the movement of the T-segment from one side of the G-gate to the other, where it can then interact with the C-gate. Furthermore, the interaction with the C-gate may be a trigger for closing of the DNA-gate and opening of the C-gate (*Figure 6*).

The complex structures provide support for previously predicted states during the catalytic cycle by showing the existence of intermediates where the T-segment is either poised to enter or after passing the DNA gate. These predictions, including the angle between the G- and T-segments and the position of the T-segment, form the bases of many models, but have not been observed before. The findings from the structures are likely relevant to other type IIA topoisomerases, not only

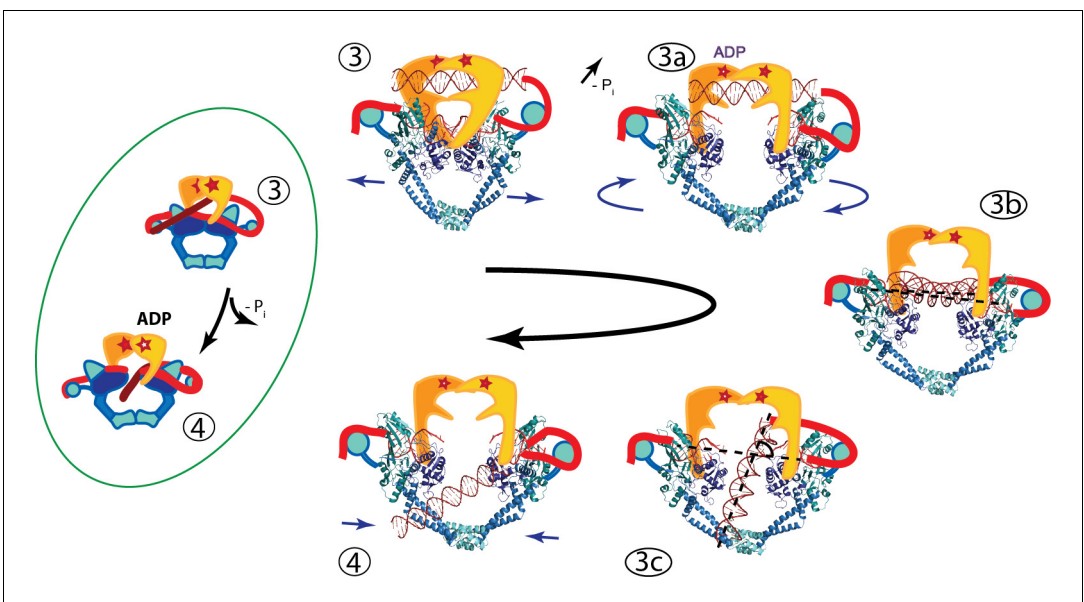

**Figure 6.** An updated mechanism of negative supercoiling by DNA gyrase. The structures of GyrA in complex with a putative T-segment leads us to propose additions to the model for gyrase supercoiling (*Figure 1*). The additions are in the steps after capture of the T-segment by GyrB leading to the passage of the T-segment through the G-gate (Panels 3 and 4 from *Figure 1*, for reference shown on the left of the figure encircled in green). Furthermore, the structures show the angle between the G- and T-segments, which are in excellent agreement with previous predictions. (3) T-segment DNA is captured by the closing of the N-gate in GyrB. 3a) The DNA-gate opens and GyrA adopts a conformation similar to the one in the Tetrahedral dimer; the subunits adjust their relative orientation to facilitate the capture and guidance of the T-segment through the DNA-gate. GyrB and GyrA rotate in a synergistic manner. (3b) GyrA and the T-segment with DNA above the DNA-gate. Both GyrA and GyrB are opened, the DNA-gate is separated, and the T-segment is poised for passing through the gate. (3c) GyrA and the T-segment with DNA below the DNA-gate. The T-segment has passed through the DNA-gate. After T-segment passage GyrA starts to rotate back with GyrB rotating in a concerted manner. In panels 3b and 3c the helical axes of the G- and T-segments are shown with dotted lines. (4) Subsequently the GyrA subunits can move back to the closed DNA-gate conformation and GyrB moves with them. Opening of the DNA-gate as described is illustrated in *Figure 6—Video 1*, which shows a morph from the closed conformation (PDB ID 4Z2C) to the Tetrahedral complex dimer to the Dihedral complex dimer.

DOI: https://doi.org/10.7554/eLife.41215.019

The following video is available for figure 6:

**Figure 6—video 1.** The movie shows a morph from the GyrA dimer in the closed conformation (PDB ID 4Z2C) to the Dihedral complex dimer to the Tetrahedral complex dimer and back to the closed conformation.

DOI: https://doi.org/10.7554/eLife.41215.020

gyrases, as they explain the manner in which DNA strands pass through the gates, a feature of both negative and positive supercoiling. In addition, our structures show that the opening of the gates is not accomplished by simple rigid body motion of two monomer subunits, but that rearrangements of the subunit domains are also needed. This inherent plasticity of GyrA may be needed for the interactions that guide the DNA through the openings in the oligomer, but also for the sensing of the DNA during the supercoiling cycle. Despite the large number of structures now available for different type II topoisomerases, it is clear that additional structures are needed showing snapshots of protein/DNA interactions throughout the catalytic cycle in order to obtain a more complete understanding of how these remarkable molecular machines perform their function.

# Materials and methods

**Key resources table**

| Reagent type (species) or resource | Designation | Source or reference | Identifiers |
|---|---|---|---|
| Gene (*Streptococcus pneumoniae*) | *gyrA*, gyrase A subunit | ATCC | 700669DQ |
| Gene (*Streptococcus pneumoniae*) | *gyrB*, gyrase B subunit | ATCC | 700669DQ |
| Strain, strain background (*E. coli*) | BL21-DE3 | Novagen | Catalogue # 1 69450 |
| Recombinant DNA reagent | pMCSG7 vector | PMID:12071693 | |
| Software, algorithm | leginon | PMID:3684574 | |
| Software, algorithm | Imagic | *van Heel and Keegstra, 1981* IMAGIC: A fast, flexible and friendly image analysis software system Ultramicroscopy 7: 113–130. | |
| Software, algorithm | tigris | tigris.sourceforge.net | |
| Software, algorithm | Motioncor2 | PMID: 28250466 | |
| Software, algorithm | mag_distortion_corr | PMID: 26278979 | |
| Software, algorithm | Relion 1.4 | PMID: 23000701 | |
| Software, algorithm | Relion 2.1 | PMID: 27845625 | |
| Software, algorithm | Relion 3 | PMID: 30412051 | |
| Software, algorithm | cryoSPARC | PMID: 28165473 | |
| Software, algorithm | cisTEM | PMID: 29513216 | |
| Software, algorithm | ctffind4 | PMID: 26278980 | |
| Software, algorithm | gctf | PMID: 26592709 | |
| Software, algorithm | REFMAC5 | PMID: 15299926 | |
| Software, algorithm | PHENIX | PMID: 20124702 | |
| Software, algorithm | MDFF | PMID:18462672 | |
| Software, algorithm | COOT | PMID: 20383002 | |
| Software, algorithm | Chimera | PMID: 15264254 | |
| Software, algorithm | PYMOL | The PyMOL Molecular Graphics System. Schrödinger, LLC | |
| Software, algorithm | superpose | PMID: 15572779 | |
| Software, algorithm | ConSurf server | PMID: 15980475 | |
| Software, algorithm | SPIDER | PMID: 19180078 | |
| Software, algorithm | APBS | PMID: 11517324 | |

*Continued on next page*

*Continued*

| Reagent type (species) or resource | Designation | Source or reference | Identifiers |
|---|---|---|---|
| Software, algorithm | jigglefit | PMID: 25615868 | |
| Software, algorithm | 3DNA | PMID: 18600227 | |
| Software, algorithm | XMIPP | PMID: 15477099 | |

## Protein purification

*Streptococcus pneumoniae gyrA* and *gyrB* genes were cloned from genomes purchased from American Type Culture Collection (ATCC, Manassas, VA) and inserted into protein overexpression vector pMCSG7 (*Stols et al., 2002*). The expression vector adds 24 amino acids, including a six amino acid His-tag and a TEV cleavage site, at the N-terminus of the protein, which were retained in all purified proteins. For protein expression, gyrase A and gyrase B were separately expressed in *Escherichia coli* BL21-DE3 cells carrying the overexpression plasmid. Cell cultures were grown at 37°C until $OD_{600}$ reached 0.8–1, then cultures were chilled at 4°C and induced with 1 mM isopropyl β-D-1-thiogalactopyranoside (IPTG). After induction cells were grown overnight at 16°C. Cell cultures were spun down and pellets were resuspended in Binding Buffer (50 mM Tris HCl pH8, 300 mM NaCl, 5 mM imidazole, 5% glycerol). Cells were lysed by incubation with 0.625 mg/ml final concentration of lysozyme, followed by incubation with 0.1% final concentration of Brij-58 and sonicated in the presence of 1 mM final concentration of phenylmethylsulfonyl fluoride (PMSF). Lysed cells were spun down at 38,000 rpm in a Ti70 rotor in a Beckman Coulter Ultracentrifuge, the cleared supernatant was filtered through a 0.2 μm filter and loaded on a Ni-Sepharose 6 Fast Flow (GE Healthcare) or Ni-NTA Agarose (Qiagen) column. After loading, the column was washed with one column volume of Binding Buffer followed by four column volumes of Wash Buffer I (50 mM Tris HCl pH8, 300 mM NaCl, 20 mM imidazole, 5% glycerol) and finally four column volumes of Wash Buffer II (50 mM Tris HCl pH8, 300 mM NaCl, 35 mM imidazole, 5% glycerol). Protein was eluted with four column volumes of Elution Buffer (50 mM Tris HCl pH8, 300 mM NaCl, 300 mM imidazole, 5% glycerol). For gyrase A purification, all eluted fractions were dialysed overnight into Heparin Buffer A (50 mM Tris HCl pH 8, 150 mM NaCl, 0.5 mM ethylenediaminetetraacetic acid (EDTA), 1 mM dithiothreitol (DTT)). Protein was filtered through a 0.2 μm filter prior to loading it into a Heparin Sepharose High Performance (GE Healthcare) column. After loading, the column was washed with Heparin Buffer A until the 280 nm UV trace stabilized around the original baseline. Protein was eluted with a NaCl gradient from 0–100% of Heparin Buffer B (50 mM Tris HCl pH 8, 1 M NaCl, 0.5 mM EDTA, 1 mM DTT). Fractions containing gyrase A were pulled together and dialyzed overnight into S300 High Salt Buffer (50 mM TrisHCl pH8, 600 mM KCl, 1 mM EDTA). For gyrase B purification, the fractions from the Ni-Sepharose 6 Fast Flow or Ni-NTA Agarose column were immediately dialyzed into S300 Low Salt Buffer (50 mM Tris HCl pH8, 150 mM KCl, 1 mM EDTA). The dialyzed fractions were concentrated to 1–2 ml volume, filtered and loaded into the HiPrep 16/60 Sephacryl S300 High Resolution column (GE Healthcare). For both proteins, after the last column the peak fractions were pooled together, concentrated, and stored frozen at −80°C.

## Complex purification

For complex formation, a 44-mer oligonucleotide with the same gyrase binding sequence as determined in DNAse protection assays (*Fisher et al., 1981*) (5′-TCGCGACGCGAGGCTGGATGGCC TTCCCCATTATGATTCTTCTC-3′) was purchased from IDT (Coralville, IN). The sense and antisense oligonucleotides were annealed together in Annealing Buffer (10 mM Tris pH8, 50 mM NaCl, 1 mM EDTA) and used without further purification. The complexes with DNA were obtained by a three step reaction, including a GraFix (*Kastner et al., 2008*) final step. First, 300 μg of GyrA-ΔCTD and 228 μg GyrB (for Tetrahedral complex) or GyrA-ΔCTD only (for Dihedral complex) were incubated in Reaction Buffer (50 mM Tris HCl pH 7.5, KCl, 5 mM DTT, 5% glycerol, 10 mM SrCl₂) for 45–60 min followed by addition of 11.6 μg of 44-mer DNA and ciprofloxacin (to a final 1.5 nM concentration) and the reaction was further incubated for 45–60 min. For the Tetrahedral complex, novobiocin (to a final 0.7 nM concentration) was added to the reaction and incubated for another 45–60 min. The reactions were loaded onto a glycerol gradient prepared as follows: an equal volume of Buffer I was

overlayed on Buffer II in a centrifugation tube (Buffer I – 50 mM Hepes pH 7.5, 55 mM KCl, 10 mM SrCl2, 10% glycerol, Buffer II – 50 mM Hepes pH 7.5, 55 mM KCl, 10 mM SrCl$_2$, 30% glycerol, 0.025% glutaraldehyde). The top of the tube was sealed with parafilm and the tube was laid sideways for 2 hr to allow for buffer mixing and gradient formation and then the tube was stood upright and stored at 4°C for another hour. After that time the reactions were loaded on top of the gradient and spun at 38,800 rpm in a Ti60SW rotor in a Beckman Coulter Ultracentrifuge at 4°C overnight. Gradients were fractionated manually. The UV absorbance of each fraction was measured using a Nanodrop ND-1000 at 260 nm and 280 nm and the values were recorded. Peak UV absorbance fractions were used for further EM studies. The antibiotics and glutaraldehyde were included as the original intent was to obtain the core complex, not just GyrA oligomers. Subsequent analyses showed that GyrB was not present and that the DNA was not in the G-segment configuration. Analyses of particles formed in the absence of glutaraldehyde and antibiotics confirmed that the particles can form without any cross-linkers or antibiotics present (*Figure 2—figure supplement 1*).

## Electron microscopy
### Initial negative stain models
#### Dihedral complex
The complex from the glycerol gradient purification was deposited into G300-Cu grids (Electron Microscopy Sciences - EMS) coated with self-prepared carbon and stained with 2% uranyl acetate (EMS) using the droplet technique (*Ohi et al., 2004*). A total of 60 negative stain images were collected manually on a JEOL 1400 microscope at 120 kV using an UltraScan4000 camera at 53,571x magnification. Initial processing was done with the Xmipp software suite (*Marabini et al., 1996*; *Scheres et al., 2008*; *Sorzano et al., 2004*). 2D classification of negative stain and initial cryoEM datasets analyzed using Xmipp provided three distinct views with clear mm mirror symmetry, suggesting D2 symmetry. Based on their dimensions it was assumed that they represent front, side and top orthogonal views and an initial model was created by back projection of the three classes in Spider (*Shaikh et al., 2008*) (*Figure 2—figure supplement 7*). In parallel, a random conical tilt reconstruction (*Radermacher, 1988*; *Radermacher et al., 1986*) produced a similar volume, confirming the initial assumption.

#### Tetrahedral complex
The complex was prepared for imaging similarly to the Dihedral complex. Around 50,000 particles were collected on a FEI Spirit microscope at 120 kV using an FEI Eagle 2K × 2K CCD camera. Processing was done with Imagic (*van Heel and Keegstra, 1981*) and Tigris (tigris.sourceforge.net). Eigenimages from a multivariate statistical analysis (MSA) indicated the presence of 2- and 3-fold symmetry and 2D classification based on the MSA produced a number of characteristic classes. Independent *ab-initio* reconstructions from the classes using common line methods and imposing C2 symmetry and C3 symmetry converged to similar results which appeared to have tetrahedral symmetry. A further reconstruction imposing tetrahedral symmetry produced a volume composed of 6 dimers of GyrA in the open conformation (*Figure 2—figure supplement 7*).

## CryoEM reconstructions
### Dihedral complex
For data collection the complex was dialyzed extensively for 22 hr after the glycerol gradient purification using a Slide-A-Lyzer MINI Dialysis Unit (ThermoFisher) to remove as much glycerol as possible. 3 µl were deposited on glow-discharged carbon-coated C-flat 2/2 µm grids (EMS), incubated for 1 min, and vitrified using a Gatan CryoPlunge3 at 85% humidity and room temperature with 3 s blotting. Initial data sets collected on a JEOL 3200FS microscope at 200kV and an UltraScan4000 camera at 75,000x magnification with 2 Å pixel size were processed using Relion 1.4 (*Scheres, 2012a*; *Scheres, 2012b*) using the low-pass filtered to 60 Å negative stain model as a starting reference and served to establish the D2 symmetry of the complex. Two additional data sets were collected on a JEOL 3200FS microscope at 300kV with a K2 Summit Direct Electron Detection (DED) camera in counting mode. These datasets improved the resolution of the complex in D2 symmetry to 7.52 Å. The D2 volume calculated from these data showed well-defined extra density in the middle of the complex that corresponded to two overlapping DNA molecules due to the symmetry. Calculations

were redone without any imposed symmetry and this model reached 9.2 Å resolution. A final data set was collected in the same manner on the JEOL 3200FS microscope but with 8 e⁻/pixel/s dose, 40,323x magnification and 1.24 Å pixel size and using Leginon (*Suloway et al., 2005*). Movies were motion-corrected and dose-weighted with MotionCor2 (*Zheng et al., 2017*) and magnification anisotropy corrected with mag_distortion_correct (*Grant and Grigorieff, 2015*). CTF parameters were calculated with CTFFIND4 (*Rohou and Grigorieff, 2015*), and data processed with Relion 1.4 (*Scheres, 2012a*; *Scheres, 2012b*). This data set yielded 112,656 good particles from 718 micrographs that were used for further refinement using the best model from the previous data set low-pass filtered to 60 Å. 2D Class averages were of excellent quality (*Figure 2—figure supplement 8*). Calculations in Relion 2.1 (*Kimanius et al., 2016*) imposing D2 symmetry and using particles where the DNA contribution had been subtracted from the images gave a reconstruction to a final resolution of 5.16 Å according to the 0.143 Fourier Shell Correlation (FSC) criterion (*Rosenthal and Henderson, 2003*). Reconstructions without subtracting the DNA, even when it was masked, went to identical resolution but consistently showed weak density for the DNA at the center. Calculations with the same data set in C1 with all the particles pre-aligned using a model including the DNA molecule yielded a map to a final resolution of 7 Å, but with the DNA clearly visible in the complex. The C1 map show that the DNA complex has C2 symmetry leading to sub-classification with C2 symmetry. In C2, three classes were recognized: one with weak DNA occupancy, one with broken DNA, and one with a clear 44 bp DNA molecule. The 30,637 particles from the class showing good DNA density were used for final refinement with C2 symmetry that went to 6.35 Å resolution. Reconstructions using the cryoSPARC (*Punjani et al., 2017*) and cisTEM (*Grant et al., 2018*) software suites produced volumes of comparable resolution and quality.

A low resolution cryo-EM reconstruction of particles assembled in the absence of glutaraldehyde or antibiotics served to confirm that the complex does not require either for assembly and that the particles still contain DNA. The crosslinker-free data were collected as described above for the high resolution Dihedral complex. A total of 37,834 particles were selected after 2D classification using particles collected from 1040 micrographs and processed using Relion-3 (*Zivanov et al., 2018*). The particles were 3D subclassified into two groups, with weak and strong DNA density. The latter group consisted of 19,197 particles and was used for 3D refinement in C2 and D2 point groups, which extended to 10.3 Å and 9.9 Å resolution. As expected, the C2 reconstruction showed density for the DNA but the D2 reconstruction did not, as it is averaged out by the extra symmetry operation. Due to the limited number of particles, the resolution of the reconstruction is low and the density is not always continuous. In particular, the DNA density is missing near the center of the volume, but very clear elsewhere. To confirm the presence of the DNA, a difference map between the C2 and D2 reconstructions was calculated in Chimera (*Pettersen et al., 2004*) and shows clear density for it (*Figure 2—figure supplement 2*).

## Tetrahedral complex

An initial cryo reconstruction of the tetrahedral complex was performed as follows: dialyzed sample was deposited onto 2/2 Quantifoil grids, which had a thin layer of continuous carbon pre-floated onto them. After incubating for 1 min, the grids were plunge frozen using a Vitrobot Mark II. Images were collected using an FEI Polara instrument, with a 2K × 2K Tietz 224HD CCD camera. Images were taken at a pixel size of 1.345 Å, and immediately binned 2-fold to produce a pixel size of 2.69 Å. For each imaged location, four images were taken, with each image having an estimated exposure of 15 e⁻/ Å². Image processing was carried out using Imagic (*van Heel and Keegstra, 1981*) and the Tigris package (tigris.sourceforge.net). Each set of 4 images were aligned to the first exposure, and summed to create a high exposure stack of images. Particles were picked using a blob template with the pick-em-all program in Imagic (*van Heel and Keegstra, 1981*) and after careful manual checking of the particles a dataset of 15,043 particles remained. Two different particle stacks were then created, one cut from the sum of 4 images to create a high dose stack, and one cut from the first image only to produce a low dose stack. The particles were refined using Tigris applying tetrahedral symmetry. The dataset was split into two independently refined halves with the high dose stack being used for refinement, and the low dose stack used for the reconstructions. The resulting structure had a resolution of ~10 Å according to the 0.143 FSC criterion (*Rosenthal and Henderson, 2003*).

A second more extensive data set was collected as follows: for data collection the Tetrahedral complex was dialyzed extensively for 22 hr after the glycerol gradient purification in a Slide-A-Lyzer MINI Dialysis Unit (ThermoFisher). Two data sets were collected on an FEI Titan Krios microscope at 300 kV with 46,430x magnification, pixel size 1.04 Å and 6 e⁻/pixel/s dose for 14 s. The first data set consisted of 1944 movies, which were motion-, anisotropy-corrected, and dose-weighted using MotionCor2 (*Zheng et al., 2017*). Initial CTF parameters were obtained with CTFFIND4 (*Rohou and Grigorieff, 2015*). Subsequent calculations were done using Relion 2.1 (*Kimanius et al., 2016*), unless noted. The total number of picked particles was 469,340 with 345,485 good particles. 2D Class averages were of excellent quality (*Figure 2—figure supplement 8*). A volume calculated from this data set based on the previous volume low-pass filtered to 60 Å show the presence of density in the center, which was interpreted as DNA. Due to the presence of preferred orientations the resolution was limited to 5.8 Å. A second data set was collected in identical fashion but with a stage tilt of 25°. The original data set had shown that most particles were aligned around the 2- and 3-fold axes, suggesting that a relatively small tilt would provide many more additional orientations. The tilted data set was processed in identical fashion. The second data set provided 736,066 picked particles. After 3D classification, 363,940 particles from the tilted data set and 345,485 particles from untilted data set were merged into one combined data set and a local CTF correction with Gctf (*Zhang, 2016*) was calculated for each particle from the merged data set. After an additional round of 3D classification the merged data set consisted of 673,694 particles. Subsequent calculations show that the combination of the two data sets, tilted and untilted, was needed to attain higher resolution. Sub-classification of the particles did not produce a higher resolution reconstruction; all particles were needed to obtain the best reconstruction. A reconstruction with imposed T symmetry, with a mask for the protein, and excluding the DNA density at the center yielded an excellent map to 4.0 Å resolution according to the 0.143 FSC criterion (*Rosenthal and Henderson, 2003*). The map shows clear density for all secondary structure elements. A second reconstruction was done without imposing tetrahedral symmetry and without masking the DNA in the center. Obtained maps were at lower resolution and contained heterogeneously oriented DNA. Further 3D classification of these particles without imposing any symmetry yielded many classes all showing DNA density of similar overall shape and size, but in slightly different orientations and positions. Two classes that showed the most promising DNA density were selected and further sub-classified into 30 groups each. From these sub-classes one of them was selected from each group and used for further refinement. These maps were at a much lower resolution density, but it was apparent that all of them have a DNA-like toroidal density ring trapped in the cage.

## Model building

### Protein models

The structure of the GyrA monomer from a *S. pneumoniae* crystal structure of GyrA and GyrB with DNA (PDB ID 4Z2C) was used as a starting model. The GyrA monomer was manually placed at one monomer position in the map of the Tetrahedral complex and the placing was adjusted using the 'jiggle fit' option (*Brown et al., 2015*) in Coot (*Emsley et al., 2010*). It was clear from this rigid body placing that some areas of the model needed to be adjusted. To accomplish this, the monomer was fitted into the map using the MDFF routines (*Trabuco et al., 2008*) that are part of NAMD (*Phillips et al., 2005*) with manual movement guidance in some regions. The monomer from MDFF (*Chan et al., 2011*; *Trabuco et al., 2008*) fitted the density well, but the stereochemistry was poor. To improve the stereochemistry and the fit, several rounds of REFMAC5 (*Murshudov et al., 1997*) and PHENIX (*Adams et al., 2010*) real space refinement were done interspersed with manual rebuilding in Coot (*Emsley et al., 2010*). Secondary structure constraints were introduced into refinements in REFMAC5 based on secondary structure assignments from PDB ID 4Z2C structure through Prosmart constraints whereas in PHENIX the secondary structure was constraint by providing HELIX and SHEET definitions from the PDB file and using the option of secondary structure restrain. Care was taken to ensure that the coordinates were constrained to have acceptable stereochemistry, including Ramachandran angles. As the Tetrahedral complex map does not show well-defined density for the side chains, most of them were placed in good rotamer positions regardless of the position in the map. Once good stereochemistry was obtained all other monomers were placed by symmetry expansion from the first monomer without further refinement. The Dihedral

complex was built in a similar manner starting from the dimer coordinates of the Tetrahedral complex.

## Protein/DNA complexes

The C2 map of the Dihedral complex showed clear density for the DNA, which runs along a diagonal in the center of the complex. A 44-mer B-DNA molecule with the appropriate sequence was fitted manually into the density in Coot (*Emsley et al., 2010*) and the fit was improved using MDFF (*Chan et al., 2011*; *Trabuco et al., 2008*). The protein coordinates from the D2 protein complex were used and adjusted by 'jigglefit' in Coot (*Emsley et al., 2010*). Further geometry refinement was performed in REFMAC5 (*Murshudov et al., 1997*) for the DNA whereas the protein coordinates were not further refined. The density for the DNA in the C1 map of the Tetrahedral complex obtained using all the particles was ambiguous and it was not clear how to build it in the map. Sub-classification produced several maps that show better DNA density, albeit calculated with a limited number of particles. Two maps were selected for model building, as described above. To fit the DNA into these maps, the density of the DNA was carved out of the map in Chimera (*Pettersen et al., 2004*) and a DNA oligonucleotide was fitted with MDFF (*Trabuco et al., 2008*) into one of the maps. These coordinates were used as a starting point to fit DNA into the other map in MDFF (*Trabuco et al., 2008*). Following MDFF (*Trabuco et al., 2008*) fitting, the DNA coordinates were combined with protein coordinates from the Tetrahedral complex and fitted again using MDFF. The stereochemistry was improved using REFMAC5 (*Murshudov et al., 1997*) and PHENIX (*Adams et al., 2010*) as described above, yet some protein-DNA clashes are still present.

The coordinates of the model all have excellent stereochemistry and fit the density well. For the Tetrahedral complex monomer the root mean square deviation (rmsd) is 0.006 Å and 0.851° for bond lengths and bond angles, respectively with 91.3% of the residues in the favored part of the Ramachandran plot and 99.3% of residues with favored rotamers. For the Dihedral complex dimer solved with D2 imposed symmetry rmsd values for bond length and bond angle are 0.005 Å and 0.888° respectively, with 92% of the residues in the favored part of the Ramachandran plot and 99.7% of residues with favored rotamers. Methionine 99 in chain A has a cis conformation.

## Model superpositions, comparisons, and figures

Comparisons of the different models were all done using CCP4 Superpose (*Krissinel and Henrick, 2004*). Boundaries for domains were defined as in *Figure 2*. When superposing structures from different organisms, the superposition was based on secondary structure matching (SSM) as implemented in Superpose (*Krissinel and Henrick, 2004*). To compare closed and open dimers, one monomer from an open dimer was superposed on the corresponding monomer of a closed dimer and the rotation angle to superpose the non-superposed monomers on each other was measured. In addition, for every domain in the superposed monomers the rotation angle needed to superpose the individual domains was measured.

Figures were drawn with Pymol ('The PyMOL Molecular Graphics System,") and Chimera (*Pettersen et al., 2004*). Angles for domain and subunit superpositions were calculated and drawn with Pymol draw_rotation_axis. Angles between the G- and T-segments of the DNA were calculated using X3DNA (*Lu and Olson, 2003*; *Lu and Olson, 2008*) and the DNA axis was plotted in Pymol. Conservation data were generated using the ConSurf server (*Ashkenazy et al., 2010*; *Landau et al., 2005*) and visualized in Pymol. Interactions between protein subunits in the Tetrahedral complex and protein subunits and DNA in the Dihedral complexes with DNA were calculated using Monster (*Salerno et al., 2004*).

## Acknowledgements

We thank members of the Mondragón and Rosenthal laboratories for discussions and assistance. We thank Andrea Nans and Jonathan Remis for help with data collection. Research was supported by the NIH (AM, grants R01-GM051350 and R35-GM118108) and by the Francis Crick Institute (PBR), which receives its core funding from Cancer Research UK (FC001143), the UK Medical Research Council (FC001143), and the Wellcome Trust (FC001143). KS was supported by the Northwestern University Molecular Biophysics Training Program. We acknowledge the help from the Northwestern University Structural Facility and the Francis Crick Institute Structural Biology Science

Technology Platform. Support from the RH Lurie Comprehensive Cancer Center of Northwestern University to the Structural Biology Facility is acknowledged. The Gatan K2 DDE at Northwestern University was purchased with funds provided by the Chicago Biomedical Consortium with support from the Searle Funds at The Chicago Community Trust.

## Additional information

### Funding

| Funder | Grant reference number | Author |
|--------|------------------------|--------|
| Wellcome Trust | FC001143 | Peter B Rosenthal |
| Cancer Research UK | FC001143 | Peter B Rosenthal |
| Medical Research Council | FC001143 | Peter B Rosenthal |
| National Institutes of Health | R01-GM051350 | Alfonso Mondragon |
| National Institutes of Health | R35-GM118108 | Alfonso Mondragon |

The funders had no role in study design, data collection and interpretation, or the decision to submit the work for publication.

### Author contributions

Katarzyna M Soczek, Software, Formal analysis, Validation, Investigation, Writing—original draft, Writing—review and editing; Tim Grant, Software, Formal analysis, Investigation, Methodology, Writing—review and editing; Peter B Rosenthal, Conceptualization, Resources, Software, Supervision, Validation, Investigation, Methodology, Writing—original draft, Writing—review and editing; Alfonso Mondragón, Conceptualization, Software, Formal analysis, Supervision, Funding acquisition, Investigation, Methodology, Writing—original draft, Project administration, Writing—review and editing

### Author ORCIDs

Katarzyna M Soczek  http://orcid.org/0000-0003-3803-6079
Tim Grant  http://orcid.org/0000-0002-4855-8703
Peter B Rosenthal  http://orcid.org/0000-0002-0387-2862
Alfonso Mondragón  http://orcid.org/0000-0002-0423-6323

### Decision letter and Author response
Decision letter https://doi.org/10.7554/eLife.41215.041
Author response https://doi.org/10.7554/eLife.41215.042

## Additional files

### Supplementary files
• Transparent reporting form
DOI: https://doi.org/10.7554/eLife.41215.021

### Data availability
Coordinates and EM maps were deposited in the PDB and EMDB with accession codes: PDB entry ID 6N1R and EMDB entry ID EMD-9318, PDB entry ID 6N1Q and EMDB entry ID EMD-9317, and PDB entry ID 6N1P and EMDB entry ID EMD-9316.

The following datasets were generated:

| Author(s) | Year | Dataset title | Dataset URL | Database and Identifier |
|-----------|------|---------------|-------------|-------------------------|
| Soczek KM, Grant T, Rosenthal PB, Mondragon A | 2018 | Tetrahedral oligomeric complex of GyrA N-terminal fragment, solved by cryoEM in tetrahedral symmetry | https://www.rcsb.org/structure/6N1R | Protein Data Bank, 6N1R |

| Soczek KM, Grant T, Rosenthal PB, Mondragon A | 2018 | Tetrahedral oligomeric complex of GyrA N-terminal fragment, solved by cryoEM in tetrahedral symmetry | https://www.ebi.ac.uk/pdbe/emdb/EMD-9318 | Electron Microscopy Data Bank, EMD-9318 |
|---|---|---|---|---|
| Soczek KM, Grant T, Rosenthal PB, Mondragon A | 2018 | Dihedral oligomeric complex of GyrA N-terminal fragment, solved by cryoEM in D2 symmetry | https://www.rcsb.org/structure/6N1Q | Protein Data Bank, 6N1Q |
| Soczek KM, Grant T, Rosenthal PB, Mondragon A | 2018 | Dihedral oligomeric complex of GyrA N-terminal fragment, solved by cryoEM in D2 symmetry | https://www.ebi.ac.uk/pdbe/emdb/EMD-9317 | Electron Microscopy Data Bank, EMD-9317 |
| Soczek KM, Grant T, Rosenthal PB, Mondragon A | 2018 | Dihedral oligomeric complex of GyrA N-terminal fragment with DNA, solved by cryoEM in C2 symmetry | https://www.rcsb.org/structure/6N1P | Protein Data Bank, 6N1P |
| Soczek KM, Grant T, Rosenthal PB, Mondragon A | 2018 | Dihedral oligomeric complex of GyrA N-terminal fragment with DNA, solved by cryoEM in C2 symmetry | https://www.ebi.ac.uk/pdbe/emdb/EMD-9316 | Electron Microscopy Data Bank, EMD-9316 |

The following previously published datasets were used:

| Author(s) | Year | Dataset title | Dataset URL | Database and Identifier |
|---|---|---|---|---|
| Laponogov I, Veselkov DA, Pan X-S, Selvarajah J, Crevel IM-T, Fisher LM, Sanderson MR | 2016 | Quinolone(Moxifloxacin)-DNA cleavage complex of gyrase from S. pneumoniae | https://www.rcsb.org/structure/4Z2C | Protein Data Bank, 4Z2C |
| Rudolph MG, Klostermeier D | 2013 | Structural plasticity of the Bacillus subtilis GyrA homodimer | https://www.rcsb.org/structure/4DDQ | Protein Data Bank, 4DDQ |
| Laponogov I, Sohi MK, Veselkov DA, Pan X-S, Sawhney R, Thompson AW, McAuley KE, Fisher LM, Sanderson MR | 2009 | Structural insight into the quinolone-DNA cleavage complex of type IIA topoisomerases | https://www.rcsb.org/structure/3FOF | Protein Data Bank, 3FOF |

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
