## [Decision Letter]

Thank you for submitting your article "CryoEM structures of open dimers of Gyrase A in complex with DNA illuminate mechanism of strand passage" for consideration by *eLife*. Your article has been reviewed by three peer reviewers, and the evaluation has been overseen by Andrea Musacchio as Reviewing Editor and Senior Editor. The following individuals involved in review of your submission have agreed to reveal their identity: Aneel Aggarwal (Reviewer #2) and Neil Osheroff (Reviewer #3).

The reviewers have discussed the reviews with one another and I have drafted this decision to help you prepare a revised submission.

Summary:

Mondragon and coworkers used cryoEM to characterize open dimers of GyrA in complex with DNA. Results provide insight into the DNA strand passage reaction catalyzed by the bacterial enzyme. They also suggest that the opening of the protein gates in gyrase is not accomplished by simple rigid body motion of the two protomer subunits. Rather, rearrangements of the subunit domains are required as well. Thus, there is an inherent plasticity to the type II enzyme. Overall, the study appears to be well done and conclusions are justified. All three reviewers agree that the paper makes substantial contributions to the field of topoisomerase structural biology, as the authors observe enzyme states not yet captured. Thus, the work is timely and will be of interest to readers of *eLife*.

Essential revisions:

1) The authors show in a supplemental figure the dihedral complex (four dimers) form in the absence of a cross linking agent. This result is used to suggest that the complex is not merely a cross-linking artefact, and is physiologically relevant. I think this is a valid argument. But then, it would be important to demonstrate the DNA binding mode in the non-cross-linked sample is the same as the cross-linked one. The presented raw image and 2D averages do not reveal how DNA binds in the complex. Because the non-cross-linked particles look good on negative stain EM images, it should be feasible for the authors to compute a medium to low resolution cryo-EM map for the sample. Because DNA is wide and relatively easy to visualize (scatters electron strongly), a 10 – 15 Å resolution map will provide a good sense of whether the DNA path in the non-cross-linked sample is the same as in their higher resolution map as described in the manuscript – where cross linking was used. Overall, this is a straightforward experiment that should not take a lot of time/effort.

2) The authors have made a well-reasoned argument that the DNA poses in dihedral complex resemble the anticipated manner how the T-segment DNA passes the G-gate. However, they should also discuss the potential caveats of their structure. In the physiological conditions, the G-gate should be a DNA gate, formed by the two cleaved ends of the G-segment DNA on top of GyrA dimer. In the reported dihedral oligomer, no DNA corresponding to the G-segment was observed. Therefore, the T-segment-like DNA in the oligomer is passing through a protein gate (formed by the two WHDs of the GyrA dimer), not a DNA gate. Furthermore, in the physiological setting, the T-segment DNA is initially admitted through the N-gate in the GyrB dimer located on top of GyrA dimer, and at the point of passing through the G-gate, the T-segment should be coordinated by GyrB on the top surface, and by GyrA and G-DNA on the bottom surface. However, in the current dihedral oligomer, both top and bottom surfaces of the T-segment DNA are surrounded by the GyrA dimers, and GyrB is absent in the structure. Therefore, the T-segment is not in a native-like environment. These two points should be noted in Discussion.

3) Gyrase requires the C-terminal domain of the GyrA subunit in order to properly wrap the DNA around itself as a prelude to supercoiling. The constructs used by the authors lacked that domain. Although I believe that the results of the work are still valid, the authors really should emphasize this issue and discuss its potential impact on their conclusions.

4) In general, the G-segment in crystal structures of type II topoisomerases (including gyrase) contains a substantial bend. The cryoEM structures presented in the current study do not show that DNA bending. The authors should address this issue in order to help bridge the compatibility of EM structures and crystal structures.

---

## [Author Response]

Essential revisions:1) The authors show in a supplemental figure the dihedral complex (four dimers) form in the absence of a cross linking agent. This result is used to suggest that the complex is not merely a cross-linking artefact, and is physiologically relevant. I think this is a valid argument. But then, it would be important to demonstrate the DNA binding mode in the non-cross-linked sample is the same as the cross-linked one. The presented raw image and 2D averages do not reveal how DNA binds in the complex. Because the non-cross-linked particles look good on negative stain EM images, it should be feasible for the authors to compute a medium to low resolution cryo-EM map for the sample. Because DNA is wide and relatively easy to visualize (scatters electron strongly), a 10 – 15 Å resolution map will provide a good sense of whether the DNA path in the non-cross-linked sample is the same as in their higher resolution map as described in the manuscript – where cross linking was used. Overall, this is a straightforward experiment that should not take a lot of time/effort.

We agree that it is important to confirm that the DNA is present in the non-crosslinked sample and follows the same path. We collected a data set from a non-cross-linked sampled and did a reconstruction, as suggested by the reviewers. The reconstruction does indeed show the DNA is present and follows the same path. The experiment was challenging, particularly with time constraints, but we have obtained a map from many fewer particles than we used to calculate our high-resolution structure but which nevertheless extends to 10.3 Å resolution showing the DNA path and supporting our strategy of using the cross-linked specimen for high resolution studies. We have added a paragraph in the Materials and methods detailing how the map was calculated. We also expanded Figure 2—figure supplement 1 to include a panel containing a difference map showing the presence of DNA together with a difference map from the original, higher resolution data set to show that a difference map calculation does show the DNA. In addition, we added a sentence in the Results section, where we describe the DNA complex, which addresses the fact that the complex does not need a crosslinker for formation:

“In addition, a low resolution cryoEM reconstruction of the Dihedral complex in the absence of antibiotics or crosslinkers shows the presence of DNA with the same path, further confirming that the DNA in the complex is not a result of crosslinking. (Figure 2—figure supplement 1B).”

2) The authors have made a well-reasoned argument that the DNA poses in dihedral complex resemble the anticipated manner how the T-segment DNA passes the G-gate. However, they should also discuss the potential caveats of their structure. In the physiological conditions, the G-gate should be a DNA gate, formed by the two cleaved ends of the G-segment DNA on top of GyrA dimer. In the reported dihedral oligomer, no DNA corresponding to the G-segment was observed. Therefore, the T-segment-like DNA in the oligomer is passing through a protein gate (formed by the two WHDs of the GyrA dimer), not a DNA gate. Furthermore, in the physiological setting, the T-segment DNA is initially admitted through the N-gate in the GyrB dimer located on top of GyrA dimer, and at the point of passing through the G-gate, the T-segment should be coordinated by GyrB on the top surface, and by GyrA and G-DNA on the bottom surface. However, in the current dihedral oligomer, both top and bottom surfaces of the T-segment DNA are surrounded by the GyrA dimers, and GyrB is absent in the structure. Therefore, the T-segment is not in a native-like environment. These two points should be noted in Discussion.

We agree that it is important to discuss potential caveats. We have added the following sentences in the Discussionto clarify the point, as requested by the reviewers:

“The absence of the G-segment in the complexes means the T-segment is passing through a gate formed only by protein, rather than DNA held apart by the protein. However, the T-segment in our complexes is compatible with other structures, such as the human topoisomerase II/DNA structure (Chen et al., 2018) that include the G-segment and polypeptide regions equivalent to portions of GyrB, supporting our conclusion that the complexes reveal bona fide strand passage intermediates.”

3) Gyrase requires the C-terminal domain of the GyrA subunit in order to properly wrap the DNA around itself as a prelude to supercoiling. The constructs used by the authors lacked that domain. Although I believe that the results of the work are still valid, the authors really should emphasize this issue and discuss its potential impact on their conclusions.

We have added the following lines in the Discussion section to address the concerns:

“GyrA-ΔCTD lacks the C-terminal domain that is essential for DNA supercoiling by gyrase, and this may in principle affect the way GyrA interacts with DNA. […] In this regard, the structures may capture a general arrangement common to all type IIA topoisomerases.”

4) In general, the G-segment in crystal structures of type II topoisomerases (including gyrase) contains a substantial bend. The cryoEM structures presented in the current study do not show that DNA bending. The authors should address this issue in order to help bridge the compatibility of EM structures and crystal structures.

Our structures do not contain the G-segment, only the T-segment. The G-segment shown in the figures comes from crystal structures, all of which show a bent G-segment. To make this clearer, we have added the following sentences in the Discussion section:

“In both of these states DNA sits within a region containing many positively charged and conserved amino acids (Figure 4—figure supplement 1) and is relatively straight, unlike G-segment DNA, which is tightly bent. […] The absence of strong protein/T-segment interactions may be needed to allow easy transport of the Tsegment through the gate, resulting in a less constrained DNA that adopts locally a linear conformation.”